# MRG15 alternative splicing regulates CDK1 transcriptional activity in mouse cell senescence and myocardial regeneration
Yuan Zhang[1,2,3,4], Huayu Wang[1,4], Fang Li[1,4], Hui Dai[1] & Ye Zhang ®[1] ✉

MRG15, a chromatin remodeling protein, plays a pivotal role in cellular senescence and proliferation. However, the precise roles and mechanisms of MRG15 in aging regulation remain unclear. Our research elucidates the distinct functions of MRG15's splice variants in aging. We find that MRG15L, contrary to the previously assumed MRG15S, accumulates with advancing age. Using histone peptide binding assays and protein interaction analysis, we demonstrate that MRG15L exhibits reduced affinity for histone H4 acetylation sites, thereby weakening CDK1 regulation, leading to G2/M phase arrest and promoting cellular senescence. During postnatal cardiac development, MRG15L expression increases and is linked to reduced regenerative capacity. Moreover, targeted knockout of MRG15L in mice enhances cardiac repair and regeneration following myocardial ischemia-reperfusion injury. These findings highlight MRG15L as a promising therapeutic target for age-related diseases, revealing its critical role in modulating aging pathways through alternative splicing.

The study of aging is a long-standing scientific pursuit, with researchers striving to elucidate the complex mechanisms that contribute to the gradual decline of organisms over time. In recent years, epigenetics has emerged as a pivotal field in aging research, suggesting that alterations in the epigenome can significantly influence cellular senescence and organismal lifespan[1,2]. Epigenetics refers to a series of reversible and heritable mechanisms that regulate gene expression and chromatin structure without altering the underlying DNA sequence. These mechanisms influence cell fate and function, and are stably transmitted during cell division[3]. Epigenetic regulation reflects both genetic background and environmental inputs, serving as a critical interface between genotype and phenotype, and between intrinsic and extrinsic factors. Core epigenetic mechanisms include DNA methylation, histone modifications, chromatin remodeling, and noncoding RNA-mediated regulation. These processes also play essential roles in the onset and progression of various age-related phenotypes[2–4].

MRG15 (MORF-related gene on chromosome 15), a chromatin remodeling factor, has been implicated in various cellular processes, including histone acetyltransferase, histone deacetylation[5], and chromatin remodeling[6]. While previous studies have suggested a link between MRG15 and aging, direct evidence of its role in senescence and the underlying mechanisms remain limited[7,8]. As a gene repeatedly implicated in aging[9], the knockdown of MRG15 unsurprisingly induces cellular senescence. Whether it is the result of intergenerational aging or the promotion of aging through overexpression, the effects are not very obvious[10,11]. Importantly, MRG15 is known to exist in two alternatively spliced isoforms—MRG15S (short form) and MRG15L (long form)[12]. Compared to MRG15S, the mouse MRG15L isoform contains an additional 39-amino-acid insertion within the Chromo domain, located immediately after lysine at position 51. Although this insertion preserves the open reading frame, it may disrupt the three-dimensional structure of the Chromo domain[12,13]. However, the functional significance of this splicing event in the context of aging has not been well characterized.

The development of CRISPR-Cas9 technology[14] has provided a powerful tool for exploring the epigenome and identifying factors that influence aging. In this study, we used a CRISPR-Cas9-based epigenetic screen in replicative senescent cells and identified MRG15 as a critical regulator of cellular aging. Among its two splice variants, MRG15L—but not MRG15S—was found to promote senescence. MRG15L expression increases during senescence, while MRG15S decreases. Deletion of MRG15L in mouse embryonic fibroblasts (MEFs) reduced *p16* levels and β-galactosidase staining, confirming its pro-senescence role. Single-cell sequencing revealed that CDK1 expression is reduced in senescent cells. MRG15S

[1]Department of Biochemistry and Molecular Biology, Institute of Basic Medical Sciences, Chinese Academy of Medical Sciences & Peking Union Medical College, Beijing, China. [2]Faculty of Hepato-Pancreato-Biliary Surgery, the First Medical Center, Chinese PLA General Hospital, Institute of Hepatobiliary Surgery of Chinese PLA, Key Laboratory of Digital Hepatobiliary Surgery, PLA, Beijing, China. [3]Department of Neurosurgery, Peking Union Medical College Hospital, Chinese Academy of Medical Sciences and Peking Union Medical College, Beijing, China. [4]These authors contributed equally: Yuan Zhang, Huayu Wang, Fang Li. ✉e-mail: yezhang@ibms.pumc.edu.cn

overexpression activated CDK1, while MRG15L did not. The accumulation of MRG15L was associated with G2/M phase arrest, suggesting that its substitution for MRG15S represses CDK1 transcription and promotes cellular senescence. We also observed that MRG15L is highly expressed in tissues with low proliferative potential, such as the heart and brain, and that its expression increases with postnatal development as MRG15S declines. These findings suggest that an age-related shift from MRG15S to MRG15L contributes to cell cycle exit and terminal differentiation. To test this, we examined the effects of MRG15L in the heart and found that mice with specific knockout of MRG15L exhibited significantly enhanced repair capabilities after myocardial ischemia-reperfusion injury compared to wild-type mice. These findings suggest that MRG15L may be a key protein in halting cell proliferation during cellular senescence and terminal differentiation.

## Results
### Single-cell sequencing reveals epigenetic factors involved in senescence of mouse embryonic fibroblasts (MEFs)

To enhance the efficiency of library infection, we generated Cre-Loxp expressing Cas9 mice to obtain MEF-Cas9 cells for replicative senescence studies. During the passaging of MEF-Cas9 primary cells, we observed a significant increase in both protein and mRNA levels of *p16*, as well as an

increase in senescence-associated β-galactosidase staining, indicating that MEF cells stably expressing Cas9 can normally undergo replicative senescence with passaging (Supplementary Fig. 1A–E). Subsequently, we infected MEF-Cas9 cells with a laboratory-constructed epigenetic factor gRNA library. After the third passage (P3) of primary cells being infected with the virus, we performed single-cell transcriptome sequencing on the seventh passage (P7) cells infected with the library and control cells (Fig. 1A). The sequencing results were divided into 13 clusters (C0-12) based on the transcriptome (Fig. 1B). GO analysis revealed that clusters C4, 5, 6, 7, 9, and 10 had high expression of proliferation-related markers such as cell proliferation and cell cycle, identifying them as anti-senescence cell clusters (Fig. 1C). The expression of the senescence marker *Cdkn2a* (*p16*) and cell proliferation marker *Mki67* also supported this clustering (Supplementary Fig. 2A–D).

To better analyze senescence-related genes, we performed The Principal Component Analysis (PCA) analysis on the co-analyzed cells. The Principal Component Analysis (PCA) significantly divided the cells into two clusters (Fig. 1D), with the senescence-related genes *Cdkn2a* and *Mki67* each scoring higher in the two clusters (Fig. 1E). This allowed us to identify other senescence-related genes. We validated the mRNA expression of these significantly different genes in replicative senescence MEF cells; genes such as *Pclaf*, *Top2a*, *Smac2*, *Stmn1*, and *Hmgb2* decreased with replicative aging,

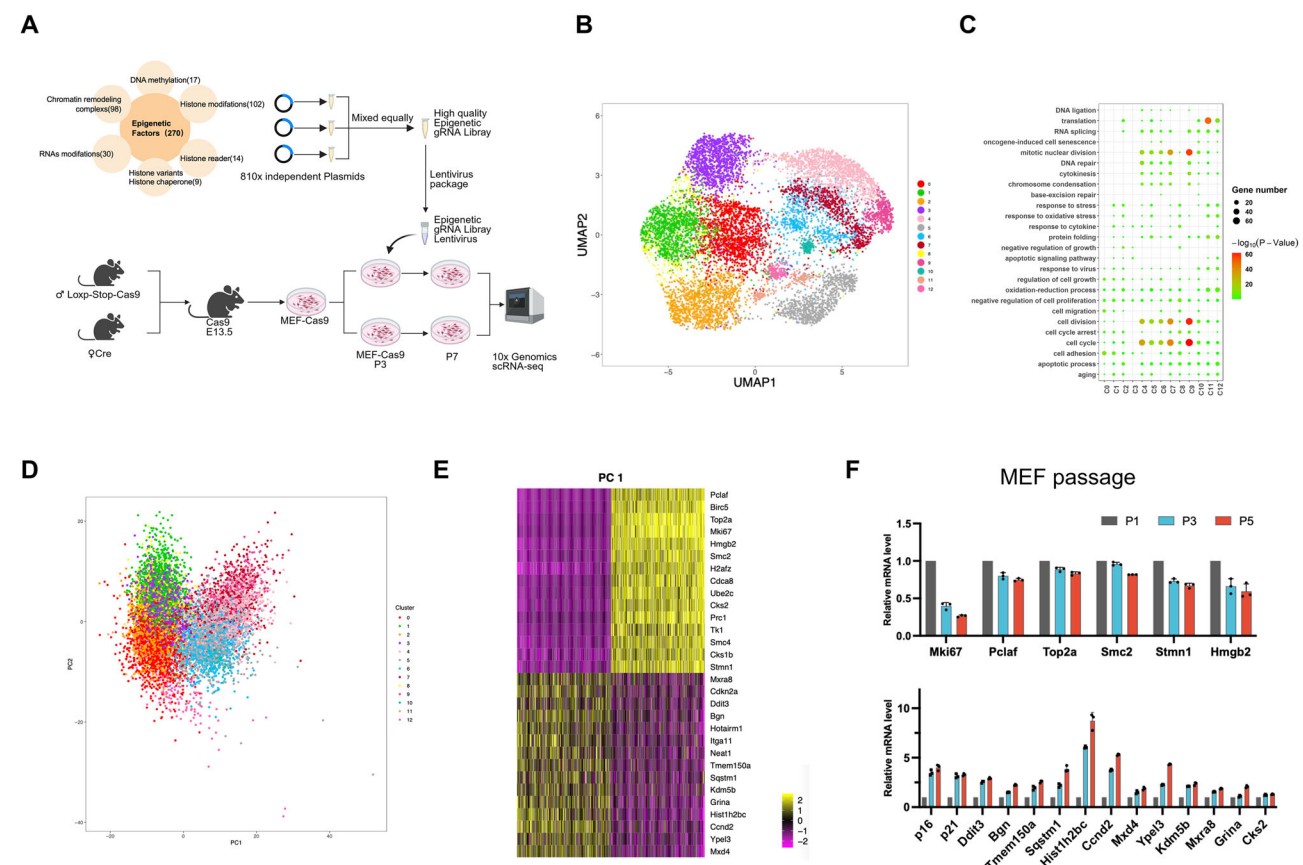

**Fig. 1 | Single-cell sequencing reveals epigenetic factors involved in senescence of mouse embryonic fibroblasts (MEFs). A** Single-cell sequencing workflow of epigenetic library screening for replicative aging genes in MEF-Cas9 cells: MEF cells expressing Cas9 were acquired from the LoxP-Cas-LoxP-Cas9 (LSL-Cas9) mice crossed with Cre mice, infected with the library at passage 3 (P3), and single-cell sequenced at passage 7 (P7) for both the control group C3 and the library group K9. **B** tSNE analysis revealed that the cells were clustered into 13 distinct groups. **C** According to the results of GO analysis, groups 4, 5, 6, 7, 9, and 10 exhibited higher expression levels of proliferation-related genes associated with cell division and the cell cycle compared to the other groups. **D** The Principal Component Analysis (PCA) shows that the overall single-cell sequencing results clearly divide the cells into two

groups: clusters 4, 5, 6, 7, 9, and 10, and clusters 0, 1, 2, 3, 8, 11, and 12. **E** PCA analysis of differential genes includes 15 increasing genes such as *ki67*, and 15 decreasing genes such as *p16*. The significant differences in *ki67* and *p16* suggest that the main basis for grouping in PCA is the degree of cell proliferation and senescence.
**F** Validation of the mRNA levels of these differential genes in P1, P3, and P5 passage senescent MEF cells shows that senescence-related genes such as *Pclaf*, *Top2a*, *Smc2*, *Stmn1*, and *Hmgb2* decrease with passage senescent, while anti-aging genes such as *Ddit3*, *Bgn*, *Timem150a*, *Sqstm1*, *Hist1h2bc*, *Ccnd2*, *Mxd4*, *Ypel3*, *Kdm5b*, *Mxra8*, *Grina*, and *Cks2* show a positive correlation with passage senescent (*n* = 3 independent experiments, data displayed as mean ± SEM). Created in BioRender. Zhang, Y. (2025) https://BioRender.com/9aja5nx.

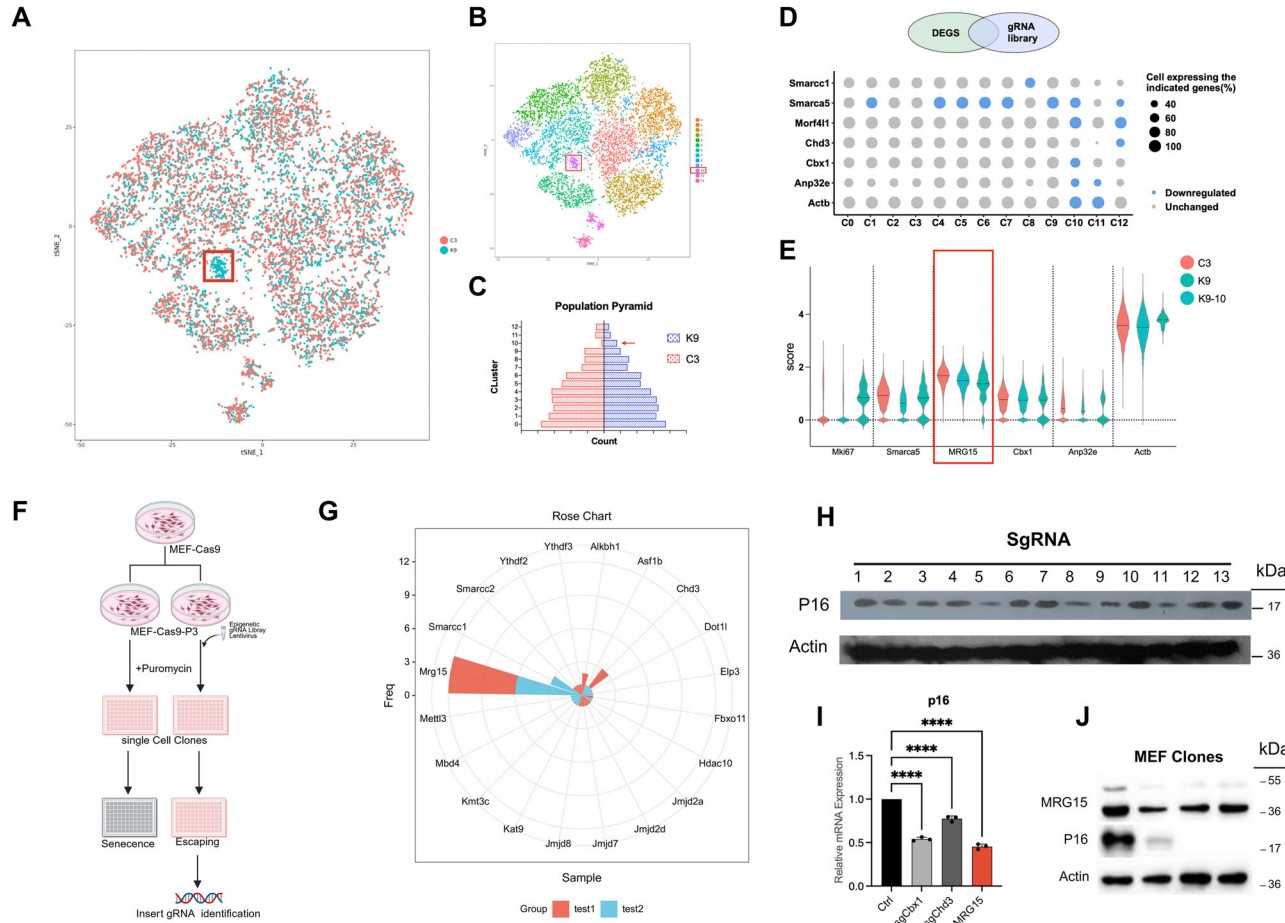

**Fig. 2 | Differential analysis of single-cell sequencing reveals MRG15 as a crucial epigenetic regulator facilitating cellular senescence escape in MEFs.**
**A** Comparison between the control group and the knockout library group reveals that the knockout group has significantly more clusters of cells. **B** The aforementioned correspond to the c10 cell cluster based on prior grouping. **C** The differential genes in the experimental group K9 compared to the C3 control group also indicate that the c10 cluster exhibits the greatest changes. **D** Transcriptome analysis of the c10 cell cluster shows a significant decrease in genes that match the library genes. Among these, epigenetic factors *Smarca5*, *Morf4l1* (*MRG15*), *Cbx1*, *Anp32*, and *Actb* are reduced, which may contribute to the formation of the c10 cluster. **E** Analysis of sequencing data for *Smarca5*, *Morf4l1* (*MRG15*), *Cbx1*, *Anp32*, and *Actb* genes in the experimental group c10 and their expression levels in the control group and other experimental groups indicates that the difference in *MRG15* is the most significant.

**F**, **G** Identification of anti-aging epigenetic genes through MEF monoclonal formation assay. After transfection with the library, MEF cells formed monoclonal colonies, and after 30 days, the majority of the colonies contained sg*Morf4l1* (*MRG15*). **H** WB validation of P16 expression levels after gene knockout. Lanes 1 to 13 correspond to Control, Smarca5, Prmt1, Cbx1, MRG15, Anp32e, Sirt1, Kat1, Kdm5b, Smarrc1, Kmt2h, CHD3 and Control. The expression levels of P16 in MEF cells significantly decreased after knocking out Prmt1, MRG15, Anp32e, and Kdm5b. **I** After knocking out the *Cbx1*, *Chd3*, and *MRG15* genes, the *p16* level in MEF cells decreased, indicating that the *p16* level was reduced following the knockout of *Cbx1*, *Chd3*, and *MRG15* ($n = 3$ independent experiments, data displayed as mean ± SEM, ****$P < 0.0001$). **J** Identification of MRG15L expression in monoclonal cell lines. Cells with low MRG15b expression also had low levels of p16. Lanes 1 to 4 correspond to MEF-WT and 3 clones of Anti-aging cell line.

while *Ddit3*, *Bgn*, *Tmem150a*, *Sqstm1*, *Hist1h2bc*, *Ccnd2*, *Mxd4*, *Ypel3*, *Kdm5b*, *Mxra8*, *Grina*, and *Cks2* increased (Fig. 1F). Many proteins, such as Top2a[15], Hmgb2[16], Sqstm1[17], Ypel3[18], and Kdm5b[19], have been reported to be associated with aging.

## Differential analysis of single-cell sequencing reveals MRG15 as a crucial epigenetic regulator facilitating cellular senescence escape in MEFs

After analyzing the comprehensive single-cell sequencing data of cellular senescence, we conducted a comparative analysis between the MEF-Cas9 cells infected with the library and the control group following serial passage. The results revealed an additional group of cells (Fig. 2A) that were not present in the control group. These cells were previously identified as cluster c10 (Fig. 2B, C), which was also recognized as an anti-aging cell cluster through GO analysis. The c10 cluster showed a significant increase in differentially expressed genes, both upregulated and downregulated, compared to other clusters, indicating that the expression profile changes were caused by the library infection (Supplementary Fig. 2E).

By comparing the differentially expressed genes with the gRNA entries in the library, we found that, in addition to the commonly upregulated *Smarca5*, the c10 cluster also showed significant differential expression of *Morf4l1* (*MRG15*), *Cbx1*, *Anp32e*, and *Actb* (Fig. 2D). Furthermore, the transcript levels of the proliferation marker *Mki67* were higher in the K9-c10 group compared to the control group, suggesting that the c10 cluster represents an anti-aging cell population. Among all the differentially expressed genes, *MRG15* showed the most significant decrease in expression, leading us to speculate that it is a key factor in the anti-aging effect of the c10 cluster (Fig. 2E).

To further validate our findings, we conducted a stringent single-colony formation assay on the MEF-Cas9 cells infected with the library. Due to the limited passaging capacity of wild-type MEF cells, they cannot form single colonies under sparse single-cell conditions. However, cells that underwent knockout in the library and exhibited anti-aging effects were able to form colonies (Fig. 2F). We identified 22 gRNAs inserted into 38 cell lines that maintained good proliferation, including *Morf4l1* (*MRG15*), *Smarcc1*, *chd3*, and *Alkbh1* (Fig. 2G). We then infected these cells with the

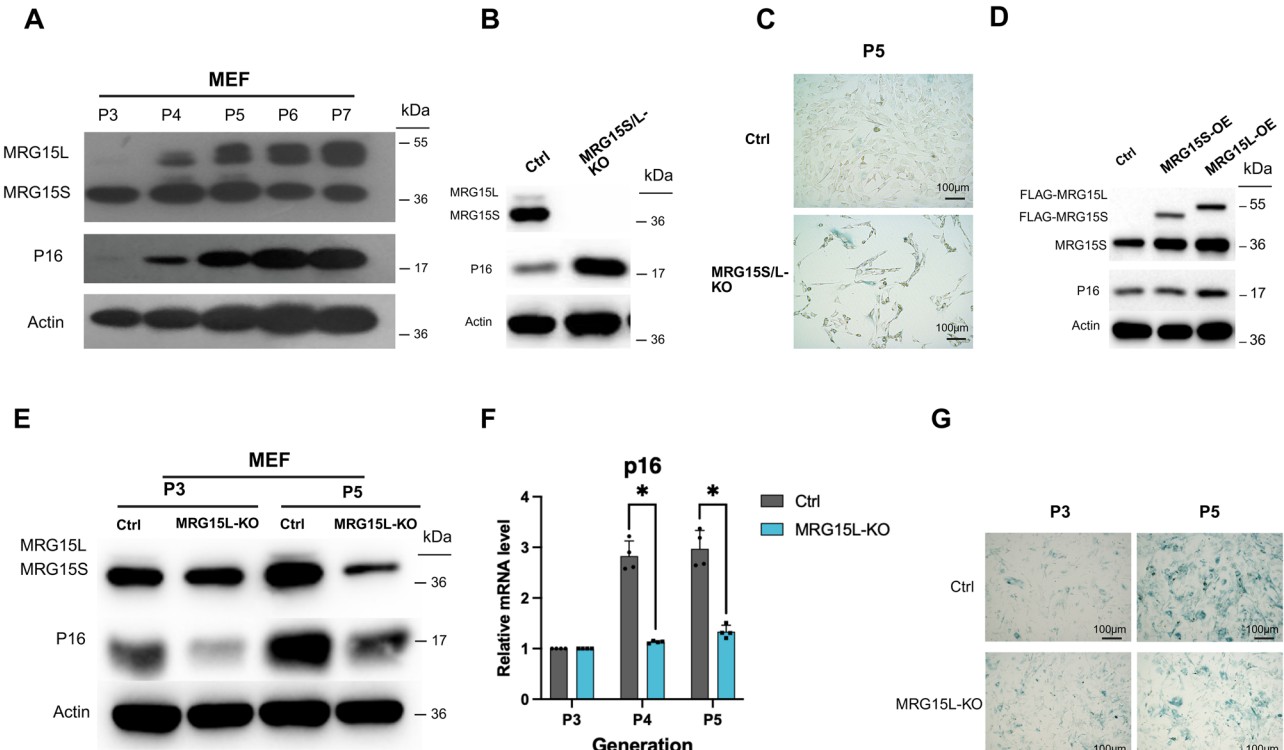

**Fig. 3 | MRG15L but not MRG15S plays a direct role in senescence regulation in MEFs. A** Detection of MRG15 expression levels in passage-senescent MEF cells from P3 to P7 generations. MRG15L expression gradually increased with each passage, while MRG15S remained unchanged. **B** Increased p16 expression levels in MEF cells with simultaneous knockout of MRG15S/L in MEF cells. **C** Increase in galactosidase staining in MEF cells with MRG15S/L knockout. **D** Overexpression of MRG15S and MRG15L in 293 T cell lines. **E, F** Reduced increase in p16 protein and mRNA expression levels across passages in MEF cells with MRG15L knockout ($n = 4$ independent experiments, data displayed as mean ± SEM, *$P < 0.05$). **G** Reduced increase in galactosidase staining in MEF cells with MRG15b knockout.

corresponding gRNAs and validated the downregulation of *p16* mRNA in cells with knockout of Cbx1, Chd3, and MRG15 using qPCR. This was associated with a significant reduction in p16 protein levels (Fig. 2H, I). Additionally, we found that the expression level of MRG15 was significantly reduced in the long-lived single-clone MEF cells compared to the control group, which was accompanied by a significant reduction in p16 expression levels (Fig. 2J). The obtained results provide confirmation that the downregulation of MRG15 can effectively postpone the replicative senescence of MEF cells.

### MRG15L but not MRG15S plays a direct role in senescence regulation in MEFs

Subsequently, we characterized the expression levels of MRG15 during replicative senescence in MEF cells. Interestingly, we observed an increase in the long variant of MRG15L during the progression of senescence in P3-P7 cells, while the short variant MRG15S remained unchanged or even decreased (Fig. 3A). This finding is consistent with our previous identification of MRG15L as a key factor in anti-aging MEF cells (Fig. 2J). Moreover, our inspection of the sgRNA constructs used in the epigenetic library screen revealed that all three target sequences were directed towards the exon 4 region specific to MRG15L, suggesting that the knockout of MRG15 in the screen was nearly equivalent to the specific knockout of MRG15L.

To further explore the roles of MRG15L and MRG15S, we designed sgRNAs targeting the shared region of MRG15S/L in the first exon and performed knockout experiments. The results showed an increase in p16 protein levels (Fig. 3B) and enhanced senescence-associated β-galactosidase staining (Fig. 3C) following the knockout of both MRG15L and MRG15S. However, when we overexpressed MRG15L and MRG15S, we found that only the overexpression of MRG15L led to an increase in p16 expression (Fig. 3D), indicating that MRG15L, rather than MRG15S, is the variant responsible for promoting senescence. In contrast, MRG15S appears to be

necessary for maintaining cell proliferation, as its knockout leads to cellular senescence.

*MRG15L* contains an additional 117 bp exon 4 compared to *MRG15S*. We designed gRNA primers targeting this exon to specifically knockout MRG15L while preserving MRG15S. The results showed that the knockout of MRG15L led to lower p16 protein levels compared to control cells (Fig. 3E, F). Additionally, during serial passaging of replicatively senescent cells, the protein and mRNA expression levels of MRG15L increased slowly, and the senescence-associated β-galactosidase staining also decreased significantly (Fig. 3G). In summary, the specific knockout of the long variant MRG15L can delay the replicative senescence of MEF cells.

### Substitution of MRG15S by MRG15L impedes the regulation of Cdk1 expression resulting in G2/M phase arrest in senescent cells

Building upon previous research, we conducted co-IP studies to investigate the protein binding partners of MRG15S and MRG15L. Our findings revealed that the short variant, MRG15S exhibits stronger binding to functional proteins such as Rb, pRb, mSin3a, and PTBP1 compared to the long variant MRG15L (Supplementary Fig. 3A, B). This observation prompted us to further explore the binding properties of MRG15S and MRG15L using histone modification chips and GST-tagged chromatin domains (Fig. 4A).

Our results showed that MRG15S has a significantly stronger binding ability to H3K36me3, H4K16ac, and H4K20me2 than MRG15L (Fig. 4B). Moreover, previous reports have documented the interaction between MRG15 (i.e., MRG15S) and H3K36me3. To validate this interaction, we performed biotin affinity precipitation experiments using biotin-labeled H3K36me3 and GST-tagged chromatin domains of MRG15S and MRG15L (Fig. 4C). The results indicated that both variants bind to H3K36me3 with similar affinity.

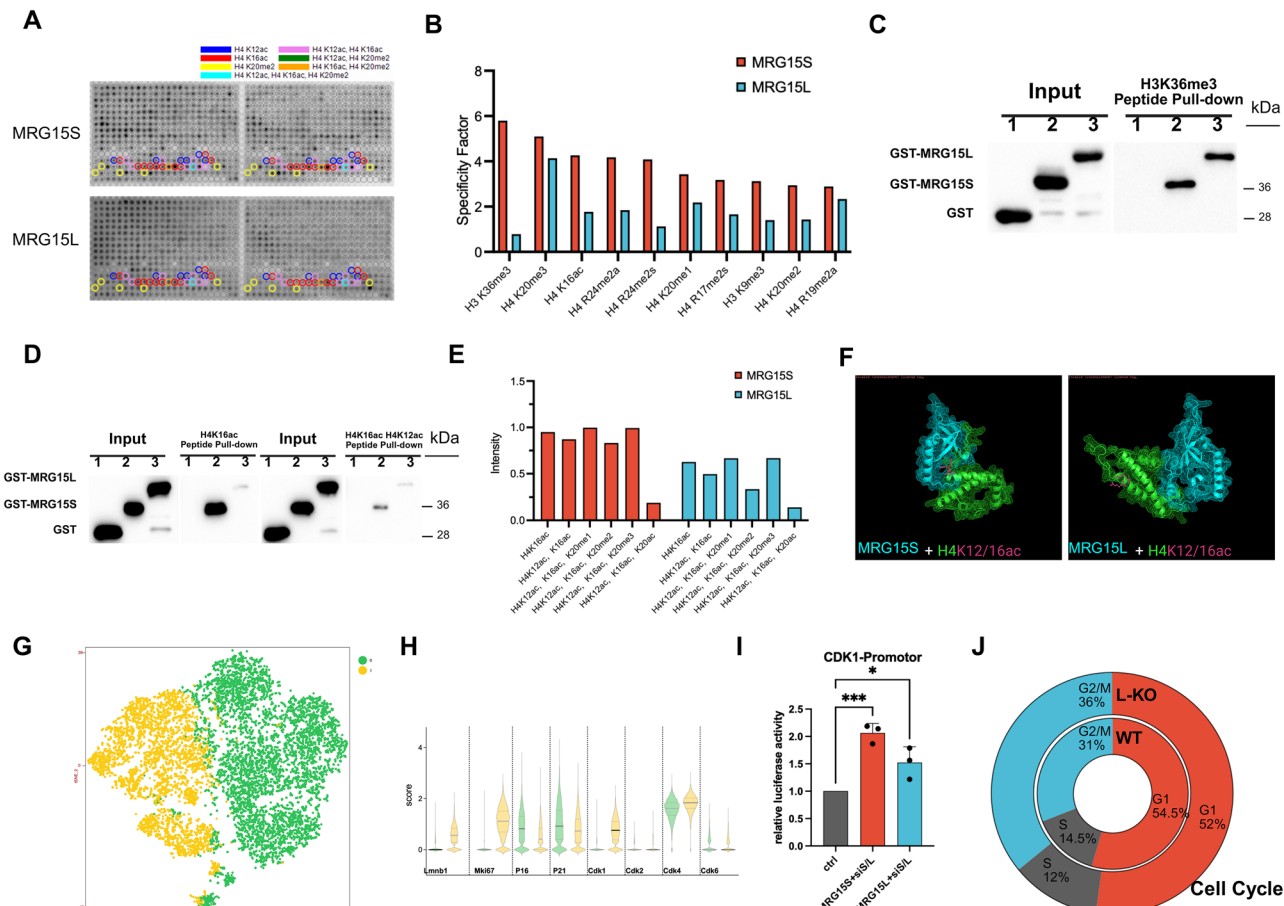

**Fig. 4 | Substitution of MRG15S by MRG15L impedes the regulation of Cdk1 expression resulting in G2/M phase arrest in senescent cells. A** Protein microarray binding results demonstrate that the MRG15S/L exhibits significant binding differences in the H4K12ac and H4K16ac regions. **B** Microarray analysis of MRG15S/L binding differences reveals distinct binding data for modifications such as H3K36me3 and H4K16ac. **C** In vitro pull-down experiments verify the H3K36me3 binding ability, with no significant difference observed between the chromodomain of MRG1S/L and H3K36me3 binding capacity. Lanes 1 to 3 correspond to GST, GST-MRG15S (N-terminal segments of MRG15-S), and GST-MRG15L (N-terminal segments of MRG15-L). **D** In vitro pull-down experiments confirm that MRG15S has a stronger binding affinity for H4K12ac and H4K16ac than MRG15L. Lanes 1 to 3 correspond to GST, GST-MRG15S (N-terminal segments of MRG15-S), and GST-MRG15L (N-terminal segments of MRG15-L). **E** The strength of H4K16ac

binding to MRG15 S/L varies, with MRG15S showing stronger binding to H4K16ac than MRG15L, particularly after H4K20me2 modification, where MRG15S' binding ability is significantly higher than that of MRG15L. **F** Structure prediction indicates that MRG15S can bind more tightly to histone H4 modified by H4K16ac and H4K12ac. **G, H** Single-cell sequencing analysis of the expression levels of cyclins in aging cell clusters reveals a significant reduction in *Cdk1* within the aging cell population, while *Cdk4/6* levels remain relatively unchanged. **I** MRG15S exhibits stronger activation capabilities for the *CDK1* promoter compared to MRG15L (*n* = 3 independent experiments, data displayed as mean ± SEM, *P < 0.05, *** *P* < 0.0005). **J** The cell cycle analysis indicates that MRG15L knockout cells have a slightly lower percentage of cells in the G1 phase, a reduced percentage in the S phase, and an increased percentage in the G2/M phase compared to wild-type cells.

Next, we examined the binding of MRG15S and MRG15L to peptides containing modifications such as H4K16ac, H4K12ac, and H4K20me2 (Supplementary Fig. 3C). Our data revealed that MRG15S has a stronger binding affinity for H4K16ac and H4K12ac than MRG15L (Fig. 4D, E). Furthermore, protein structure predictions suggest that the additional peptide segment in MRG15L alters the conformation of the CHROMO DOMAIN, thereby weakening its binding to H4K16ac and H4K12ac (Fig. 4F).

In prevous studies, we were able to observe a marked increase in MRG15 at the *cdc2* promoter. The fact that this was accompanied by an increase in acetylated histone H4, involving lysine 12 and lysine 16[20]. Meanwhile, in our analysis of single-cell transcriptome sequencing data, we identified two distinct cell populations: a senescent group (group 0) and an anti-aging group (group 1) (Fig. 4G). The expression levels of *Mki67* and *P16* were consistent with their respective group identities. Notably, we observed a significant decrease in *Cdk1* transcript levels in senescent cells, while *Cdk4/6* levels remained unchanged (Fig. 4H).

To further validate these findings, we conducted dual-luciferase reporter assays and found that MRG15S, but not MRG15L, could activate *CDK1* transcription more strongly(Fig. 4I). Additionally, cell cycle analysis revealed an increased proportion of G2/M phase cells in MRG15L-knockout MEFs compared to wild-type controls (Fig. 4J and Supplementary Fig. 5), suggesting that the substitution of MRG15L for MRG15S in replicatively senescent cells leads to decreased *CDK1* transcription activity, resulting in G2/M phase arrest. These findings imply that knocking out MRG15L can alleviate the G2/M phase arrest associated with cellular senescence.

## Knockout of MRG15L promotes repair and regeneration after myocardial ischemia-reperfusion injury

In the process of aging, MRG15L inhibits the G2/M phase of the cell cycle, leading to cellular senescence. The regulation of the cell cycle is not limited to the aging process; it also plays a significant role in cell differentiation and proliferation. We observed the expression of MRG15 variants in different

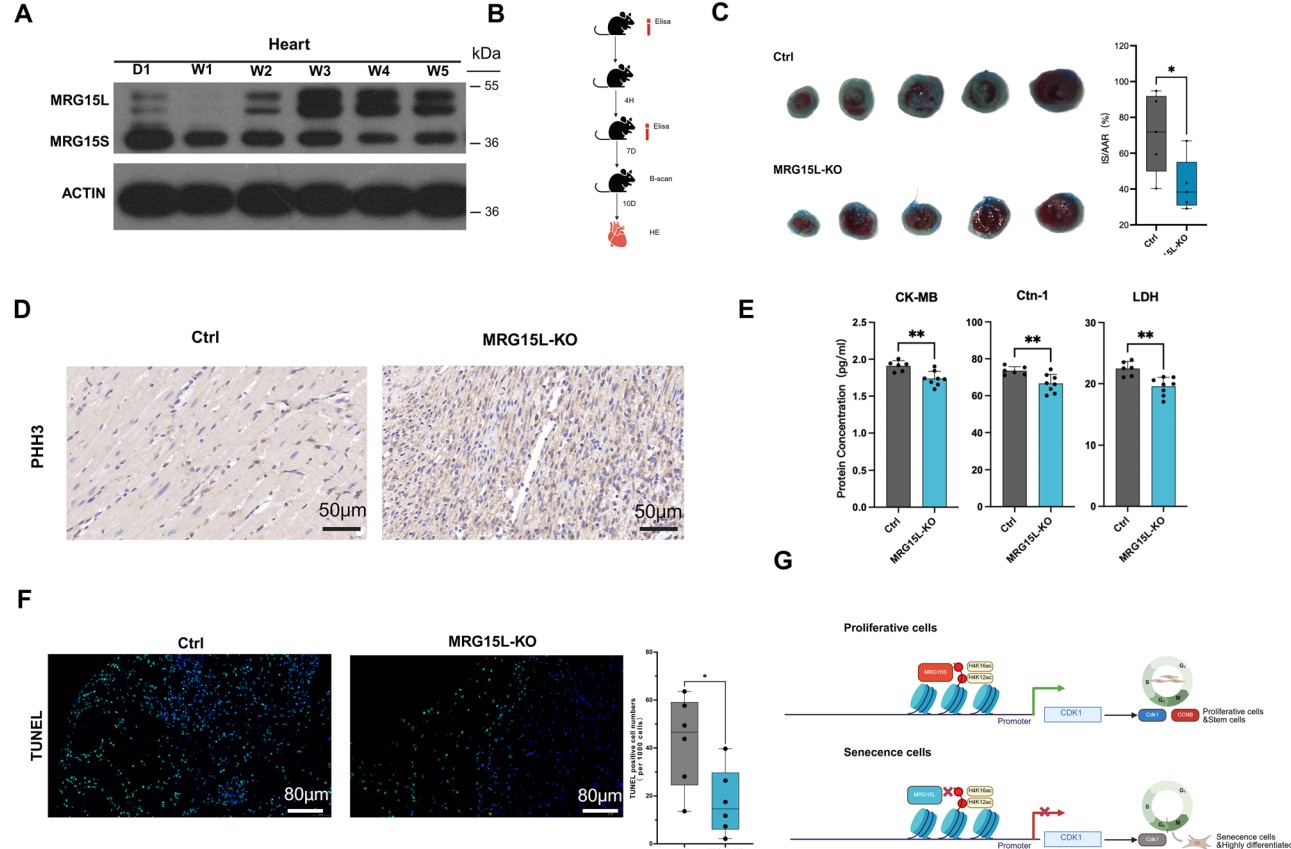

**Fig. 5 | Knockout of MRG15L promotes repair and regeneration after myocardial ischemia-reperfusion injury. A** Expression of MRG15S/L during myocardial development. From day 1 to 5 weeks of heart development, MRG15S expression decreases while MRG15L expression increases with early myocardial development. **B** Myocardial injury protocol: Blood samples were taken before injury and 4 h later for ELISA, followed by ultrasound examination after 7 days, and histological and staining studies of the myocardium after 10 days. **C** TTC and Evans Blue staining in control and MRG15L-KO groups: The infarcted area is significantly reduced in the knockout group ($n = 5$ mice per group, data displayed as mean ± SEM,*$P < 0.05$). **D** Compared to the wild-type group, PHH3 levels increase after MRG15L knockdown, indicating partial regeneration capacity in the heart. **E** Injury markers CK-

MB, cTnI, and LDH are significantly decreased in the MRG15L-KO group ($n = 6$ mice per group, data displayed as mean ± SEM, **$P < 0.005$). **F** TUNEL staining is reduced after knockdown, indicating that cell apoptosis is also alleviated following injury ($n = 6$ mice per group, data displayed as mean ± SEM, *$P < 0.05$). **G** Schematic diagram of MRG15 regulation: In proliferating cells, MRG15 tends to be MRG15S, which can bind to the region of histone H4 acetylation and activate CDK1 to promote cell passage through the G2/M phase. In senescent or terminally differentiated cells, MRG15L replaces MRG15S, thereby weakening its binding ability to Histone H4 acetylation and leading to a lack of CDK1, which causes cell cycle arrest. Created in BioRender. Zhang, Y. (2025) https://BioRender.com/9aja5nx.

mouse tissues. Our comprehensive analysis revealed that the short variant, MRG15S, is predominantly expressed in the embryo, spleen, and lungs, while the long variant, MRG15L, exhibits high expression levels in the heart, liver, kidney, and muscles. Both variants show relatively high expression levels in the cerebrum and cerebellum (Supplementary Fig. 4A), which is further supported by PCR analysis of their RNA (Supplementary Fig. 4B), indicating significantly higher expression in the heart and brain.

Upon investigating the expression levels of MRG15 variants in heart and brain tissues of postnatal mice from the first day after birth to five weeks (Supplementary Fig. 4C), we observed a gradual decrease in the expression of the short variant MRG15S, whereas the expression of the long variant MRG15L progressively increased (Fig. 5A). Additionally, we verified the protein levels of MRG15S/L variants in several cell lines using Western blot (WB) analysis and found that, in most of the stable cell lines available in our laboratory, MRG15S is predominantly expressed over its long variant MRG15L. (Supplementary Fig. 4D).

It has been reported that mouse cardiomyocytes still possess a strong regenerative capacity around one week after birth[21], and the MRG15L variant appears in the myocardium after the first week. We speculate that there may be a close relationship between MRG15L and the regeneration capacity of mouse myocardium. Studies have shown that overexpression of cyclins[22], especially cyclin B1-CDC2 (CDK1), can promote the proliferation

of cardiomyocytes[23]. Previous studies have shown that mice with a complete knockout of MRG15 lead to embryonic lethality, so we constructed mice with a specific knockout of MRG15L.

Following myocardial ischemia-reperfusion injury, TTC and Evans Blue staining of heart tissue from wild-type (WT) and MRG15L knockout (KO) mice revealed a significant reduction in infarct size in the knockout group, decreasing from approximately 70% to 40% (Fig. 5B, C and Supplementary Fig. 6A). Furthermore, the expression of the proliferation marker PHH3 was markedly elevated in MRG15L-KO mice (Fig. 5D and Supplementary Fig. 6C), and their cardiac ejection fraction exhibited a slight improvement compared to WT controls (Supplementary Fig. 6B). Elisa detection of heart injury risk factors CK-MB, cTnI and LDH also showed better results in the MRG15L knockout group than in the control group (Fig. 5E and Supplementary Fig. 6D). Additionally, apoptosis analysis revealed a lower apoptotic rate in the knockout group (Fig. 5F). Notably, P16 immunohistochemical staining at the infarct border zone showed a significant attenuation of cellular senescence in MRG15L-KO mice compared to WT controls (Supplementary Fig. 6E). In conclusion, the elimination of MRG15L led to enhancements in the repair capabilities and physiological markers of mouse myocardial cells following ischemia-reperfusion injury. Consequently, MRG15L appears to function in the suppression of the cell cycle and cell proliferation during differentiation, akin to its involvement in

the aging process. The targeted ablation of MRG15L further has the potential to reinstitute a measure of regenerative capacity to the mouse myocardium.

## Discussion

The research on the relationship between the *MRG15* gene family and aging has a long history, almost concurrent with the famous aging gene *Age-1*[24]. In 1983, Smith J.R. and Pereira-Smith, O.M. discovered the predecessor of the MRG15 (Morf4l1) family, MORF4, in their study of immortalized cell lines[25]. Initially thought to be an inactive pseudogene[26], MORF4 sparked ongoing research into the connection between the MRG family and aging, shifting the focus from MORF4 to the more widely expressed MRG family protein Morf4l1, also known as MRG15[27]. The study of MRG15 continues to evolve, with researchers identifying other members and variants of the MRG family, such as MRGX (Morf4l2), and exploring the mechanisms by which MRG15 may influence aging. These include its interactions with Rb protein and b-Myb[28], its potential role in histone acetylation complexes like NuA4[29] and deacetylation complexes like HADC[30], and its binding to histone modifications such as H3K36me3 through its Chromo Domain[31]. Additionally, there is ongoing research into the potential roles of MRG15 in neural development[32] and breast cancer[33].

Some studies have made rapid progress, such as the successful resolution of the crystal structure of MRG15's Chromo and MRG domains[6,31,34]. However, other studies, like those on the relationship between MRG15 and aging, have progressed more slowly. Despite MRG15's consistent appearance in aging screens[9] and its classification as an aging-related gene in databases, research has stalled due to several factors: the continuous increase in protein and mRNA levels during cellular senescence[35], and the embryonic lethality resulting from its complete knockout[10].

Our research suggests that MRG15S/L variants should be studied separately. Protein variants also have precedents in aging control. Progeria is caused by a point mutation in the 11th exon of the LMNA gene (GGC-GGT), which creates a new splicing site, resulting in the removal of 50 carboxyl-terminal amino acid residues from prelamin A peptide chain, thus producing progerin[36]. However, unlike progerin caused by splicing errors, the regulation between MRG15 variants is apparently more precise and complex. *MRG15S* may be an important gene for regulating cell proliferation and development, as its knockout leads to growth retardation and embryonic developmental failure. In contrast, *MRG15L* acts as a regulatory gene for aging, with its expression increasing during terminal differentiation or senescence in tissues such as the heart and brain, leading to an arrest in cell division. In senescent cells, knocking out MRG15L allows them to escape replicative senescence. Our studies on cells with a specific knockout of MRG15L show delayed replicative senescence, but this alone does not achieve the immortalization observed in previous gRNA screens. MRG15L appears to act as a gatekeeper for proliferation. While its knockout does not directly induce widespread and rapid proliferation, it may prime certain cells to respond more effectively to proliferative stimuli, thereby playing a crucial role in regeneration and repair.

Although this study focuses on the regulation of MRG15S/L variants in cell replication aging, and myocardial development processes, we have also observed the regulation of MRG15S/L in the process of neural development. Another protein variant that has sparked intense discussion in neuronal reprogramming in recent years is Ptbp1 and Ptbp2. Researchers believe that orderly knocking down Ptbp1 and Ptbp2 in astrocytes can induce their transdifferentiation into neuronal cells[37,38]. Similar to MRG15, the simultaneous presence of Ptbp1/2 does lead to cell apoptosis. Previous studies on MRG15 have also indicated its interaction with Ptbp2, jointly regulating the splicing of Tnp2 pre-mRNA in round spermatocytes[39]. We have simultaneously detected MRG15S and L variants in mouse brain tissue by Western blotting and have also discovered the appearance and increase of MRG15L during the postnatal neurodevelopment process in mice. Just like the widespread distribution of Ptbp1 and the specific distribution of Ptbp2 in neurons and testes, we also speculate that MRG15S may be distributed in astrocytes believed to have proliferative potential, and MRG15L may be distributed in mature differentiated neurons. Previous studies by Martin M. Matzuk and others have shown that MRG15 gradually increases during sperm maturation, with our speculation in tissue development suggesting the increase is attributed to the MRG15L variant. Our previous experiments have shown a stronger interaction between MRG15S and PTBP1, making the interaction between MRG15L and Ptbp2 in testicular tissue more plausible. Different combinations of MRG15S-Ptbp1 and MRG15L-Ptbp2 variants may play important roles in determining the fate of neurons and astrocytes, similar to their role in testicular maturation. Furthermore, PTBP1 controls the splicing of DPF2, with its inhibition of exon 7 of Dpf2 producing DPF2-S transcripts in early development. In the neural differentiation stage, loss of PTBP1 leads to the inclusion of exon 7, resulting in DPF2-L transcripts that are typically expressed in flattened, aging-like cells[40]. Just as previous studies have shown that high expression of RBM4 always produces short splicing, while low expression produces long splicing[41], MRG15L, PTBP2, and DPF2-L seem to share a common splicing regulation system. Of course, these speculations require further evidence as the study of regulation between protein variants is a massive undertaking. We hope to see more research focusing on epigenetic regulation, as well as developmental aging and other regulatory aspects.

Carlos López-Otín and Guido Kroemer et al. believe that aging is driven by hallmarks fulfilling the following three premises: (1) their age-associated manifestation, (2) the acceleration of aging by experimentally accentuating them, and (3) the opportunity to decelerate, stop, or reverse aging by therapeutic interventions on them[2]. The expression of MRG15L increases with the replicative senescence of MEF cells, while knocking out MRG15L can delay cellular senescence, and overexpressing MRG15L can also increase the expression of P16. MRG15L meets the characteristics of these aging hallmarks, and it is thus presumed that MRG15L, rather than the commonly studied MRG15S, is a cause of aging. *Ddit3*, *Bgn*, *Tmem150a*, *Sqstm1*, *Hist1h2bc*, *Ccnd2*, *Mxd4*, *Ypel3*, *Kdm5b*, *Mxra8*, and *Grina*, which were previously identified and preliminarily validated by single-cell sequencing, all exhibit increased expression with the replicative senescence of MEF cells and can serve as potential aging biomarkers for further investigation.

During the development of the heart and nervous system, we observed a phenomenon similar to aging, where MRG15L replaces MRG15S. Although it is widely accepted that cellular aging differs from terminal differentiation[42], evidence of aging and proliferation regulation in neuronal differentiation has also been observed[40]. LMNA protein variants have been linked to premature aging syndromes and cardiomyopathy[43]. Additionally, the clearance of senescent cells in the mouse heart has been shown to promote cardiac remodeling and regeneration[44], suggesting that in non-proliferative tissues such as myocardium and neurons, aging and differentiation share certain similar mechanisms. This overlap is one of the reasons why distinguishing between aging and differentiation has been challenging. Our research has found that the replacement of MRG15S with MRG15L leads to weakened binding with histone H4 acetylation in the promoter region of *CDK1*, resulting in cell cycle arrest at the G2/M phase. Indeed, both histone H4 acetylation and the cell cycle play a broader role in regulating various physiological processes[45,46]. In this study, knocking out MRG15L in aging and myocardial injury cells was able to restore some of their proliferative capacity, suggesting that the regulation of MRG15L and MRG15S variants may influence a broad range of proliferation-related processes, with significant implications for both physiological and pathological conditions in organisms.

## Methods

### Isolation and culture of mouse embryonic fibroblasts (MEFs)

Mouse embryonic fibroblasts (MEFs) were isolated from E13.5 embryos of C57BL/6 J mice following ethical guidelines. Embryos were collected aseptically and minced into small pieces after removing heads, limbs, and visceral organs. The tissue fragments were incubated with 0.25% trypsin-EDTA (M&C Gene Technology, catalog no. CC017) at 37 °C for 15 min, then dissociated cells were pelleted and resuspended in DMEM (M&C Gene

Technology, catalog no. CM15019) with 10% FBS (Gibco) and 1% penicillin-streptomycin (abm, Cat. G255). MEFs were plated and maintained at 37 °C with 5% $CO_2$. After 24 h, non-adherent cells were removed, and the remaining cells were designated as passage 0 (P0). Cells were passaged every three days at a density of $3 \times 10^5$ cells per 10 cm dish (3T1 passaging).

## Real-time qPCR and analysis

Total RNA was isolated from cells using the FastPure Cell/Tissue Total RNA Isolation Kit V2 (Vazyme, Cat. RC112-01). Reverse transcription to complementary DNA was performed using the All-In-One 5× RT MasterMix (abm, Cat. G592). Real-time qPCR was conducted using BlasTaq™ 2× qPCR MasterMix (abm, Cat. G5891) on a CFX 100 system (Bio-Rad Laboratories), with Gapdh mRNA used as the internal control. Relative gene expression levels were calculated using the $2^{-\Delta\Delta Ct}$ method. Primer sequences are listed in Supplementary Table 1. All qPCR experiments were performed in three independent experiments. Data are presented as mean ± SEM, and statistical significance was determined using GraphPad Prism 9.

## Generation of MEF-Cas9 cells

To generate MEF-Cas9 cells, male mice carrying a LoxP-stop-LoxP-Cas9 (LSL-Cas9) allele were bred with female mice expressing Cre recombinase under a tissue-specific promoter. Offspring were genotyped to identify those with successful Cre-mediated excision of the stop cassette, resulting in stable Cas9 expression. Pregnant Cre-positive females were sacrificed at embryonic day 13.5 (E13.5) to harvest embryos. Embryos were individually processed to isolate primary mouse embryonic fibroblast (MEF) cells. Each embryo's MEF cells were cultured separately, and Cas9 expression was verified by Western blot analysis, using HEK293T cells transiently transfected with Cas9 as the positive control group.

## Construction of the epigenetic factor gRNA library

The selection of epigenetic factors primarily referenced C. David Allis' second edition of "Epigenetics" and Abcam's epigenetic modifications. Based on the sgRNA sequence website (http://crispor.tefor.net), we designed and selected the top three scoring synthetic primers for each epigenetic factor. In total, 270 epigenetic factors were selected, yielding 810 pairs of gRNA primers. The full list of gRNA sequences is provided in Supplementary Data 1. Each gRNA was individually extracted from plasmids and quantified for concentration. After equal concentration mixing, they were packaged into lentiviruses. Following titration, the viruses were utilized for subsequent experiments.

## Gene overexpression, deletion, and silencing

To delete the endogenous *MRG15L* gene in MEF-Cas9 cells, we employed the CRISPR/Cas9 method by injecting guide RNA targeting the fourth exon of the *MRG15* gene into fertilized eggs of C57BL/6 J mice. Homozygous mice with the fourth exon of MRG15 knocked out, specifically eliminating the long splice variant of MRG15 protein, were identified and bred.

For the overexpression of MRG15S/L in HEK293T cells (preserved in our laboratory), we prepared plasmids pCMV-3Tag-6-M4S (for MRG15S) and pCMV-3Tag-6-M4L (for MRG15L), along with an empty vector control. The coding sequences for MRG15S and MRG15L were optimized for human expression. Plasmid DNA was diluted in Opti-MEM and mixed with a transfection reagent according to the manufacturer's instructions. Stable transfected cells were selected with 200 µg/mL hygromycin B for 2–3 weeks until hygromycin-resistant colonies were visible. Resistant colonies were picked, expanded, and validated for MRG15S/L overexpression using Western blot analysis. Validated clones were cryopreserved in DMEM with 10% FBS and 10% DMSO for future use.

For siRNA knockdown experiments, stable HEK293T cell lines expressing MRG15S/L were transfected with siRNAs (abm) targeting the endogenous *MRG15* gene. Nontargeting (scrambled) siRNA served as the negative control. After 48 h, protein extracts were isolated and subjected to immunoblot analyses. The siRNA primer sequences used in this study are summarized in the Supplementary Table 2. MRG15S/L overexpressing HEK293T cells at 50% confluency were transfected with 50 nM siRNA using Lipofectamine 2000 (Thermo Fisher Scientific, 11668019) according to the manufacturer's protocol.

## Western blotting

For western blotting, cells were lysed in RIPA buffer with protease inhibitors (Beyotime, P1045), and protein lysates were mixed with loading buffer and denatured at 100 °C for 5 min. Denatured samples were either used immediately or stored at –20 °C. Proteins were separated by SDS-PAGE using 12% resolving gels and 5% stacking gels. Gels were prepared by sequentially mixing 30% acrylamide solution, Tris buffer, 10% SDS, ultra-pure water, and finally 10% ammonium persulfate and TEMED. Proteins were electrophoresed using 1× running buffer at 80 V until entering the resolving gel, then at 120 V until the dye front reached the bottom. Proteins were transferred to nitrocellulose membranes using a wet transfer system (300 mA, 2.5 h). Membranes were rinsed with TBST, blocked with 5% non-fat milk at room temperature for 1 h, and incubated with primary antibodies diluted in 3% BSA at 4 °C overnight. After washing with TBST three times (10 min each), membranes were incubated with HRP-conjugated secondary antibodies diluted in 5% milk for 1 h at room temperature, followed by three TBST washes. Chemiluminescent detection was performed using freshly prepared ECL substrate (Thermo Fisher Scientific, 34580), and membranes were imaged using a chemiluminescence detection system (Tanon 5800 Multi). Antibodies used are listed in Supplementary Table 3.

## Co-immunoprecipitation (Co-IP)

Co-immunoprecipitation of FLAG-tagged fusion proteins was performed using EZview™ Red ANTI-FLAG® M2 Affinity Gel (Sigma, F2426). Briefly, 40 µL of well-suspended affinity gel was transferred to a pre-chilled 1.5 mL microcentrifuge tube on ice. The gel was equilibrated by washing three times with 500 µL of TBS buffer (50 mM Tris-HCl, 150 mM NaCl, pH 7.4), each time centrifuged at $8200 \times g$ for 30 s and discarding the supernatant. The prepared cell lysate containing FLAG-fusion proteins (amount depending on expression level) was then added to the gel and incubated overnight at 4 °C with gentle rotation. On the next day, the gel was centrifuged at $8200 \times g$ for 1 min to remove unbound material. The beads were washed three times with 500 µL of TBS buffer for 5 min each at 4 °C with gentle inversion. Bound proteins were eluted by adding 40 µL of 3×FLAG peptide elution buffer (Sigma, F4799) and incubating for 4 h at 4 °C. After centrifugation at $8200 \times g$ for 1 min, the supernatant was collected and used directly for subsequent Western blot analysis to assess protein–protein interactions.

## Dual-luciferase reporter assay

A firefly luciferase reporter plasmid (pGL3-basic) containing the entire ~3400 bp promoter region of *CDK1* (−3200 to +75 bp) was constructed by cloning the promoter sequence into the pGL3-Report vector. The Renilla luciferase control plasmid (pRL-TK) served as an internal control to normalize transfection efficiency. Cells were seeded in 24-well plates at a density of $1 \times 10^5$ cells per well and transfected using Lipofectamine 2000 (Thermo Fisher Scientific, 11668019) according to the manufacturer's instructions. After 24 h, cells were harvested and the luciferase assay was performed using the Luciferase Assay Kit (Promega, E1960). Firefly luciferase activity was normalized to Renilla luciferase activity to control for transfection efficiency, and relative luciferase activity was calculated by dividing the firefly luciferase signal by the Renilla luciferase signal.

## Senescence-associated β-galactosidase (SA-β-gal) staining

The X-gal staining solution was prepared with the following components: 1 mg/mL X-gal (from a 20 mg/mL stock solution dissolved in dimethylformamide and protected from light), 40 mmol/L citric acid/sodium phosphate buffer (pH 5.6), 5 mmol/L potassium ferricyanide, 5 mmol/L potassium ferrocyanide, 150 mmol/L NaCl, and 2 mmol/L $MgCl_2$. The solution was stored at 4 °C in the dark and used within 2 weeks.

MEF cells were seeded in 6-well plates at a density of $1 \times 10^5$ cells per well. After 24 h, the culture medium was removed and cells were washed twice with PBS. Cells were fixed in 1 mL of 4% paraformaldehyde in PBS for 10 min at room temperature, followed by two PBS washes. Then, 1–2 mL of X-gal staining solution was added per well. Plates were wrapped in aluminum foil to protect from light and incubated overnight at 37 °C (without $CO_2$). After 12–16 h, cells were washed twice with PBS, and stained cells were visualized under a light microscope.

### Expression and purification of GST-tagged proteins
Plasmids containing GST-tagged full-length MRG15S, MRG15L, and the N-terminal segments of MRG15S and MRG15L were transformed into *E. coli* BL21 (DE3) competent cells (Trans, CD601-02). Transformed cells were grown overnight at 37 °C in LB medium containing ampicillin (100 μg/mL), then diluted 1:100 into fresh medium and grown until the $OD_{600}$ reached 0.6-0.8. Protein expression was induced with 0.2 mM IPTG at 16 °C for 20 h. Cells were harvested and resuspended in ice-cold BC-500 buffer (20 mM pH 7.9 Tris-Cl, 500 mM NaCl, 1.5 mM MgCl2, 10% glycerol, 0.5% TritonX-100, 1 mM PMSF, 1 mM DTT), lysed by sonication, and clarified by centrifugation. The supernatant was incubated with Glutathione-agarose 4B (GE Healthcare, 17-0756-01) beads pre-equilibrated with PBS, washed with BC100 buffer (20 mM pH 7.9 Tris-Cl, 100 mM NaCl, 1.5 mM MgCl2, 10% glycerol, 0.5% TritonX-100, 1 mM PMSF, 1 mM DTT). GST-fusion proteins were eluted using 10 mM reduced glutathione in 50 mM Tris-HCl buffer (pH 8.0), then separated by SDS-PAGE and visualized by Coomassie Brilliant Blue staining.

### Histone peptide array assay
Histone peptide arrays (Active Motif, 13005) were used to assess the binding affinity of GST-tagged recombinant proteins to various histone modifications, following the manufacturer's protocol. Briefly, arrays were first placed in an appropriate incubation chamber and washed twice with PBS-T (PBS containing 0.05% Tween-20). Blocking was performed with 10 mL of 5% non-fat milk at room temperature with gentle agitation for 1 h and 20 min to reduce non-specific binding. After a single PBS-T wash, the arrays were pre-incubated with chilled pre-binding buffer (3% BSA in PBS-T) at 4 °C for 10 min.

Purified GST-tagged recombinant proteins were diluted to a final concentration of 175 nM in 4 mL of binding buffer (PBS-T containing 0.45% BSA, 0.5 mM EDTA, 0.1 mM DTT, and 10% glycerol), and applied to the array surface. The arrays were incubated overnight at 4 °C with gentle agitation. On the following day, arrays were washed three times with PBS-T for 5 min each. A primary anti-GST antibody was added and incubated overnight at 4 °C with gentle agitation. After washing three times in PBS-T, an appropriate HRP-conjugated secondary antibody was added and incubated for 1 h at room temperature, followed by three additional washes in PBS-T. Detection was performed using ECL chemiluminescence reagents (Thermo Fisher Scientific, 34580), and signal acquisition was carried out using a chemiluminescence imaging system (Tanon 5800 Multi). Data were analyzed using Array Analyze software provided with the array platform. Antibodies used are listed in Supplementary Table 3.

### Peptide pulldown assay
Four biotinylated peptides were designed and synthesized: H3K36me3, H4K16ac, H4K12acK16ac, and H4K12acK16acK20me2. Peptides (2 mg) were dissolved in 4 mL PBS. Streptavidin magnetic beads (Thermo Fisher Scientific, 11205D) were washed three times with PBS containing 0.1% TritonX-100 and then incubated with 200 μL peptide solution and 400 μL beads overnight at 4 °C with gentle rotation. After incubation, unbound peptides were removed by washing three times with PBS/0.1% TritonX-100. Peptide-bound beads were then resuspended in PBS containing 0.1% BSA and 0.02% NaN₃ and stored at 4 °C for subsequent use. For binding assays, the peptide-bead complexes were washed and incubated with purified recombinant GST-fusion proteins at 4 °C overnight with rotation. The beads were then washed three times with PBS containing 0.5% TritonX-100

and 400 mM NaCl to remove unbound proteins. After final washing, bound proteins were eluted by boiling in SDS loading buffer at 100 °C and analyzed by Western blot to assess peptide–protein interactions.

### Cell cycle analysis
Cell cycle analysis was conducted using the Cell Cycle Analysis Kit (Beyotime, C1052) following the manufacturer's protocol. MEF cells were fixed in pre-cooled 70% ethanol at 4 °C overnight, treated with RNase A, and stained with propidium iodide (PI). DNA content was measured using a CytoFLEX flow cytometer (Beckman). The proportions of cells in G1, S, and G2/M phases were determined based on PI fluorescence intensity using FlowJo software. At least three independent experiments were performed.

### Mouse myocardial ischemia-reperfusion injury model
All animal experiments were conducted in compliance with the ARRIVE guidelines. We have complied with all relevant ethical regulations for animal use. Male MRG15L knockout mice and wild-type C57BL/6 J mice (12 weeks old) were used in this study. All animals were housed under standard conditions and acclimatized for 7 days prior to experiments. Mice were anesthetized using inhaled isoflurane. A left thoracotomy was performed at the fourth intercostal space along the left sternal border to expose the heart. Myocardial ischemia was induced by ligating the left anterior descending (LAD) coronary artery with a 7-0 silk suture. After 30 min of occlusion, the ligature was released to allow reperfusion for 10 days. Successful occlusion was indicated by pallor of the distal myocardium and ST segment elevation on the ECG. Restoration of coronary blood flow was confirmed by return of myocardial color and a reduction of ST elevation by more than 50%. Infarct size was determined by TTC and Evans Blue staining, and echocardiographic assessment of cardiac function was performed 7 days post-MI/R injury.

### Detection of myocardium infarct size
After reperfusion, the heart was excised, sectioned, and stained with TTC and Evans Blue to visualize infarcted myocardium. Infarct size was quantified by calculating the percentage of the ischemic risk area relative to the total left ventricular cross-sectional area and the percentage of the infarcted area relative to the ischemic risk area.

### Echocardiographic assessment of cardiac function
Cardiac structure and function were assessed using the Animal Ultrasound Imaging System (FUJIFILM VisualSonics Vevo 3100). Left ventricular ejection fraction (EF) and fractional shortening (FS) were calculated and recorded.

### Detection of myocardial injury biomarkers
Serum levels of cardiac Troponin I (cTnI), Creatine Kinase-MB (CK-MB), and Lactate Dehydrogenase (LDH) were measured using commercially available assay kits (Solarbio) following the manufacturer's protocols. Blood samples were collected from mice before the ischemia-reperfusion procedure and again 4 h after reperfusion. Samples were left undisturbed at room temperature for 2 h, then centrifuged at $4500 \times g$ for 10 min to isolate serum. The absorbance was measured using a microplate reader, and biomarker concentrations were calculated based on standard curves provided in the kits.

### Immunohistochemistry (IHC)
Paraffin-embedded tissue sections were deparaffinized in xylene (twice, 15 min each), followed by a graded ethanol series (100%, 95%, 85%, and 75% ethanol, 5 min each), and rinsed in distilled water. Antigen retrieval was performed using 1× citrate buffer (pH 6.0) in a pressure cooker for 2 min after boiling. Endogenous peroxidase activity was blocked with 3% $H_2O_2$ for 20 min at room temperature, followed by three washes in PBS (5 min each). Tissue sections were then circled with a hydrophobic pen and blocked with 10% goat serum for 30 min at 37 °C. After removing excess serum, sections were incubated with primary antibodies diluted in antibody diluent at 4 °C

overnight or at 37 °C for 2 h. The next day, slides were washed in PBST and incubated with HRP-conjugated secondary antibodies for 1 h at 37 °C, followed by three PBST washes. DAB chromogen solution was added, and the reaction was monitored under a microscope until brown precipitate appeared, indicating positive staining. Sections were rinsed with water to stop the reaction. Hematoxylin counterstaining was performed for 3–5 min, followed by differentiation with 0.5% acid alcohol for 1–2 s and bluing in 0.2% ammonia water. Dehydration was performed through a graded series of ethanol and n-butanol (5 min each), followed by two xylene washes. Finally, coverslips were mounted using neutral balsam. Antibodies used are listed in Supplementary Table 3.

## TUNEL assay

Apoptosis was analyzed using the TUNEL Apoptosis Kit (PINUOFEI BIOLOGICAL, P0017). Paraffin sections were deparaffinized in xylene (5–10 min, 2–3 times), rehydrated through graded ethanol (100%, 85%, 75%), and rinsed in distilled water. After PBS rinse, a hydrophobic barrier was drawn around the tissue with a PAP pen. To permeabilize, samples were incubated with Proteinase K working solution (1:99 in PBS) at 37 °C for 20 min and washed three times with PBS. Slides were then incubated with 50 μL of TUNEL reaction mixture (prepared according to manufacturer's instructions) at 37 °C for 2–3 h in a humidified, dark chamber. After washing, nuclei were counterstained with 50 μL DAPI for 2–3 min at room temperature in the dark. Slides were rinsed again in PBS and mounted using antifade mounting medium. Samples were immediately examined under a fluorescence or bright-field microscope.

## Statistics and reproducibility

Statistical analyses were conducted using GraphPad Prism 9 software. Comparisons between two groups were evaluated using unpaired two-tailed t-tests. Data are presented as mean ± SEM. A $p$-value less than 0.05 was considered statistically significant. Sample sizes are indicated in figure legends, and all experiments were repeated at least three times with biologically independent samples, unless otherwise specified.

## Ethics

We have complied with all relevant ethical regulations for animal use. All animal procedures were conducted in accordance with the guidelines set forth by the Beijing Municipal Committee for the Use and Care of Laboratory Animals and complied with all relevant ethical standards. The animal experiments in this study were conducted at Beijing Medconn Biotechnology Co., Ltd. (Animal Use License: SYXK (Jing) 2020-0050). All experimental protocols were approved by the Laboratory Animal Welfare and Ethics Committee of Beijing Medconn Biotechnology Co., Ltd. under the approval number MDKN-2023-049.

## Reporting summary

Further information on research design is available in the Nature Portfolio Reporting Summary linked to this article.

## Data availability

All data supporting the findings of this study are available within the article and its Supplementary Information files. Source data underlying the figures and tables are provided in the Supplementary Data 1. Sequencing data have been deposited in Science Data Bank (DOI: 10.57760/sciencedb.25314)[47]. All plasmids are available at Science Data Bank under https://doi.org/10.57760/sciencedb.25259[48]. Original blot/gel images for all the experiments presented in the manuscript can be found in Supplementary Fig. 7. The source data behind the graphs in the paper are provided in Supplementary Data 2. Additional data and materials are available from the corresponding author upon reasonable request.

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

## Acknowledgements

We gratefully acknowledge Dr. Yufei Shen for her pioneering work in establishing the Laboratory of Chromatin and Epigenetic Regulation at PUMC and CAMS in 1984. This work was supported by the National Natural Science Foundation of China (91519301) and the CAMS Initiative for Innovative Medicine (2016-I2M-3-002).

## Author contributions

Ye.Z. (Ye Zhang), Yu.Z. (Yuan Zhang), H.W. and F.L. jointly designed the experiments. Yu.Z. performed CRISPR-Cas9 screening, single-cell RNA sequencing data analysis, and generation of MEF-Cas9 cells. H.W. and F.L. conducted all qPCR, Western blotting, Co-IP, peptide pull-down assays, and related data analysis. H.W. additionally carried out the histone peptide array assays, SA-β-gal staining, and CDK1-related experiments. F.L. performed the overexpression experiments. F.L., H.W., and Yu.Z. jointly completed all other experimental work and data interpretation. H.D. provided technical assistance and team coordination throughout the study. The manuscript was reviewed and edited by Yu.Z., H.W., F.L., and Ye.Z. Ye.Z. provided funding, coordinated the research, and supervised the project.

## Competing interests

The authors declare no competing interests.
