## [Transparent Peer Review file · Communications Biology]

MRG15 alternative splicing regulates CDK1 transcriptional activity in mouse cell senescence and myocardial regeneration

Corresponding Author: Professor Ye Zhang

Version 0:

Reviewer comments:

Reviewer #1

(Remarks to the Author)

This is an interesting and well-designed study describing the modulation of cellular senescence and myocardial regeneration by MRG15. The study elucidates the distinct roles of two MRG15 splice variants, MRG15L (variant b) and MRG15S (variant a), in the context of aging. The authors demonstrate that MRG15L plays a particularly significant role in promoting cellular senescence both in vitro and in vivo.

Overall, this study provides a solid foundation for understanding the impact of chromatin remodeling on the mechanisms of cellular senescence as well as tissue repair and regeneration during aging. However, it lacks several critical pieces of information that the authors should address to enhance the study's overall impact.

The authors show that MRG15L reduces the expression of senescence biomarkers such as p16 and beta-gal in MEFs, suggesting that the knockout of MRG15L could promote repair and regeneration by reducing senescent cells in an in vivo mouse model following myocardial ischemia-reperfusion injury. However, the authors do not provide evidence demonstrating the improvement of these processes in cardiomyocytes. To properly evaluate senescence and DNA damage markers, the authors should isolate RNA or protein specifically from cardiomyocytes. As is well-known, the heart contains a heterogeneous cell population, including endothelial cells, smooth muscle cells, and cardiomyocytes. Therefore, to assess cardiomyocyte-specific senescence and the aging-related secretory phenotype (SASP) via RT-PCR or Western blotting, cardiomyocytes should be isolated using established protocols (e.g., doi: 10.1152/ajpheart.00465.2021; <https://doi.org/10.1038/s42003-021-02677-y>). Alternatively, senescence could be assessed through immunohistochemistry or immunofluorescence, using methods described in the literature (e.g., doi: 10.1016/j.mad.2022.111740).

Additionally, the study does not address the impact of oxidative stress and apoptosis in cardiac tissue from MRG15L-KO mice, which would be crucial for a comprehensive analysis.

Minor comments:

- SUPPLEMENTARY TABLE 3: This table is truncated in the main text. Please correct this issue.
- Scale bars: Scale bars are missing in the figures. Ensure they are included and explicitly mentioned in the figure legends.
- Statistics: Statistical methods and data should be included in the main text or detailed in the figure legends.

Reviewer #2

(Remarks to the Author)

This is a well-written manuscript on the extremely timely topic.

In my opinion, several comments should be considered.

As for in vitro part:

1. Explain why the cells reach so early replicative senescence (and refer to the Hayflick limit values).
2. In this regard, can telomere length be measured?

While the information obtained in vitro show valuable insight in the senescent cell biology, the in vivo experiments are not convincing.

1. As for the injury, the samples from non-infarcted animals (blood and myocardial tissue) must be included.
2. Echocardiography: the data in Figure S5 show no difference, contrary to what is stated in the results. Please include echocardiography traces on which the quantification was done. Show also the control (non-infarcted heart).
3. The level of PHH3 labeling shown in Figure 5D is impressive. Thus, it is critical to show high magnification images of proliferating cells with co-labeling with the markers of cardiomyocytes and vascular cells.
4. Cell growth-quantification: provide the fraction of PHH3-labelled cardiomyocytes of cardiomyocytes (following immunolabeling of cardiomyocytes in tissue sections)
5. Cell death decrease at 10 days. It is necessary to show high magnification images to see which myocardial cells are TUNEL-positive.
6. Indicate the portion of myocardium (which zone) within which the quantification was done.

Of curiosity, can we see the indicators of senescence in myocardial tissue of MRG15L-KO? (in this case it is plausible to consider stress-induced senescence instead of replicative senescence)

Minor points:

In the legend to Figure 5, panel F, why use "necrosis"?

Is the legend to Figure S3, panel B correct?

In the legend to Figure S4, panel A, what is "central liver"?

Line 53 "pro-aging" might sound better.

Version 1:

Reviewer comments:

Reviewer #1

(Remarks to the Author)

The authors have addressed all my concerns

Reviewer #2

(Remarks to the Author)

The issues were addressed. Thank you for the clear response.

Response to Reviewer #1

We sincerely appreciate the reviewer's positive evaluation of our manuscript and their constructive feedback, which has helped us improve the quality of our work.

Q1) Regarding the assessment of cardiomyocyte-specific senescence and the aging-related secretory phenotype (SASP):

*Thank you for your insightful suggestion. We acknowledge the heterogeneity of cardiac tissue, which includes endothelial cells, smooth muscle cells, and cardiomyocytes. To more accurately assess cardiomyocyte senescence, we have included **p16 and PHH3 immunohistochemical (IHC)** staining, demonstrating a significant reduction in senescent cells in MRG15L-KO mice compared to WT controls (Figure 5D, S5E). We believe that these methods, as described in the literature (e.g., doi: 10.1016/j.mad.2022.111740), provide sufficient evidence supporting our findings. Additionally, we have referenced relevant conclusions from "The clearance of senescent cells in the mouse heart also promoted cardiac remodeling and regeneration" (citation 40), which aligns with our findings. Consistent with this, our data show a significant reduction in P16 staining and an increase in PHH3-positive cardiomyocytes in MRG15L-KO mice, further supporting its role in mitigating myocardial senescence and promoting cardiac repair.*

Q2) Evaluation of Oxidative Stress and Apoptosis:

*We appreciate the reviewer's valuable input. To assess oxidative stress and apoptosis in cardiac tissue, we have included **TUNEL assay results**, which indicates a lower apoptotic rate in MRG15L-KO mice compared to controls (Figure 5F). Additionally, ELISA quantification of cardiac injury biomarkers (CK-MB, cTnI, and LDH) further supports reduced myocardial damage in the knockout group (Figure 5E, S5D), providing a more comprehensive analysis of the protective effects of MRG15L deletion.*

Q3) Table truncation issue:

*Thank you for pointing this out. We have **corrected the truncated table** in the main text.*

Q4) Scale bars in figures:

*We appreciate your attention to detail. We have **added scale bars to Figures 3C, 3G, 5C, and 5D**, and explicitly mentioned them in the respective figure legends.*

Q5) Statistical methods and data:

*Thank you for this important suggestion. We have now included the statistical methods and data in the main text and detailed them in the figure legends. Specifically, we have **added the sample size (n) and p-values to Figures 1F, 2I, 3F, 4I, and 5C, E, F**.*

Response to Reviewer #2

We deeply appreciate the reviewer's insightful comments and suggestions, which have significantly enhanced the clarity and depth of our manuscript.

Q1) Explain why the cells reach so early replicative senescence (and refer to the Hayflick limit values).

*Thank you for your insightful question. The Hayflick limit for mouse cells is approximately 12 population doublings. In our study, the cells underwent a **total of 10-11 doublings from P0 to P7**. As shown in Figure S1A, MEF-Cas9 cells exhibited a 52-fold increase from passage 2 (P2) to passage 7 (P7), corresponding to approximately 5-6 population doublings. Based on the 6-fold proliferation observed between P2 and P3, and the 36-fold expansion from P0 to P2 (equivalent to ~5 doublings), the total number of population doublings from P0 to P7 was calculated to be 10-11. The additional stress from Cas9 and passaging losses likely contributed to the cells reaching replicative senescence earlier than expected. This is consistent with the observed increase in senescence markers such as P16 and β -galactosidase.*

Q2) In this regard, can telomere length be measured?

*We sincerely thank you for your valuable suggestion. Following the reviewer's suggestion, we attempted to measure telomere length using qPCR. However, we **did not observe significant telomere shortening**. We hypothesize that this may be due to the relatively short time frame of our in vitro experiments, as telomere shortening is a gradual process that may not be as pronounced in rapid passaging conditions. While we acknowledge the relevance of telomere dynamics in replicative senescence, we focused on P16 and β -galactosidase as more immediate markers of senescence in our study.*

Q3) As for the injury, the samples from non-infarcted animals (blood and myocardial tissue) must be included.

*Thank you for raising this critical point. We have now included blood samples from both **infarcted and non-infarcted animals in our revised analysis (Figure S5D)**. However, due to technical limitations in collecting paired myocardial tissue (infarcted and non-infarcted regions) from the same mouse, we compared MRG15L-KO mice with wild-type (WT) controls to evaluate tissue-*

specific effects of MRG15L deletion. This approach ensures robust comparisons while adhering to ethical and experimental constraints.

Q4) Echocardiography: the data in Figure S5 show no difference, contrary to what is stated in the results. Please include echocardiography traces on which the quantification was done. Show also the control (non-infarcted heart).

*We appreciate the reviewer's careful evaluation of our echocardiography data. As highlighted in **Figure S5B**, while MRG15L-KO mice showed a modest trend toward improved cardiac function post-IRI, these differences were "**slightly better**" (indicated by "ns" in the updated figure). To ensure transparency, we have now included representative echocardiography traces for infarcted hearts in Figure S5B, corresponding to the quantification. However, due to experimental design constraints, echocardiography was performed only on infarcted hearts, and non-infarcted heart data were not collected in parallel. We acknowledge this as a limitation and appreciate the reviewer's suggestion for future studies.*

Q5) The level of PHH3 labeling shown in Figure 5D is impressive. Thus, it is critical to show high magnification images of proliferating cells with co-labeling with the markers of cardiomyocytes and vascular cells.

*Thank you for your valuable suggestion. We recognize the importance of confirming proliferating cell identity through high-magnification imaging and co-labeling with cardiomyocyte and vascular markers. Our **PHH3 staining (Figure 5D)** shows a significant increase in proliferating cells in MRG15L-KO mice. While we did not initially perform co-labeling, our focus was on overall myocardial proliferation. To enhance clarity, we have included higher magnification images of PHH3-positive cells in the revised figure. We acknowledge that future studies incorporating co-labeling with cardiomyocyte (e.g., cTnT) and vascular markers (e.g., CD31, α -SMA) will further refine cell-type specificity.*

Q6) Cell growth-quantification: provide the fraction of PHH3-labelled cardiomyocytes of cardiomyocytes (following immunolabeling of cardiomyocytes in tissue sections)

*We appreciate this recommendation. As requested, we have quantified the fraction of PHH3-labeled cardiomyocytes in tissue sections and included the data in **Figure S5C**.*

Q7) Cell death decrease at 10 days. It is necessary to show high magnification images to see which myocardial cells are TUNEL-positive.

*Thank you for this suggestion. We have updated **Figure 5F** with high-magnification fluorescent TUNEL images, clearly identifying TUNEL-positive cardiomyocytes. These images further confirm the reduction in apoptosis in MRG15L-KO mice compared to controls.*

Q8) Indicate the portion of myocardium (which zone) within which the quantification was done.

*We appreciate your attention to methodological detail. All quantifications (e.g., infarct size, senescence markers) were performed specifically within the **infarct border zone**, as now explicitly stated in the **Figure S5A legend** and Methods section.*

Of curiosity, can we see the indicators of senescence in myocardial tissue of MRG15L-KO? (in this case it is plausible to consider stress-induced senescence instead of replicative senescence)

*Thank you for this insightful suggestion. We have added **P16 immunohistochemistry** data from infarcted hearts in **Figure S5E**, which demonstrates significantly reduced senescence in MRG15L-KO mice compared to WT controls.*

Minor points:

- **Figure 5F legend:** “Necrosis” has been revised to “**apoptosis**.”
- **Figure S3B legend:** Corrected to “**PCMV-3tag6-MRG15S**.”
- **Figure S4A legend:** “Central liver” clarified to “**liver**.”
- **Line 53:** “aging” revised to “**pro-aging**.”

We hope that these revisions address the reviewer's concerns and improve the overall quality of our manuscript. Thank you once again for your valuable feedback